# Recent Advances in Synergistic Effect of Nanoparticles and Its Biomedical Application

**DOI:** 10.3390/ijms25063266

**Published:** 2024-03-13

**Authors:** Sitansu Sekhar Nanda, Dong Kee Yi

**Affiliations:** Department of Chemistry, Myongji University, Yongin 17058, Republic of Korea; nandasitansusekhar@gmail.com

**Keywords:** synergistic effect, nanoparticles, nanoalloy, bimetallic nanoparticles, green synthesis

## Abstract

The synergistic impact of nanomaterials is critical for novel intracellular and/or subcellular drug delivery systems of minimal toxicity. This synergism results in a fundamental bio/nano interface interaction, which is discussed in terms of nanoparticle translocation, outer wrapping, embedding, and interior cellular attachment. The morphology, size, surface area, ligand chemistry and charge of nanoparticles all play a role in translocation. In this review, we suggest a generalized mechanism to characterize the bio/nano interface, as we discuss the synergistic interaction between nanoparticles and cells, tissues, and other biological systems. Novel perceptions are reviewed regarding the ability of nanoparticles to improve hybrid nanocarriers with homogeneous structures to enhance multifunctional biomedical applications, such as bioimaging, tissue engineering, immunotherapy, and phototherapy.

## 1. Introduction

The immense beneficial impacts of science and technology on society and the general wellbeing of humankind are widely recognized. Within the field of nanotechnology, materials in a wide range of sizes and morphologies at the nanoscale have been produced. Nanoparticles (NPs) present unique properties that differ from those of their bulk counterparts due to quantum-size effects at nanometer size dimensions [1]. NPs in a variety of shapes and dimensionalities, ranging from one to three dimensions, have been extensively studied to investigate how quantum size effects impact their physical, chemical, and photothermal properties [2]. The morphology, size, and surface of metallic NPs can be uniquely tailored to specific medicinal and biological applications [3]. Their physicochemical properties change with the NP size, degree of NP concentration, etc., which may also affect their surface plasmonic characteristics [4]. Functionalized NPs have an intricate structural architecture, with a composition that can be divided up into three overlying levels: (a) the innermost core, the most fundamental part of the NP structure; (b) the shell that surrounds the core and may be adorned with ions, surfactants, polymers, and small molecules; and, finally, (c) the surface, functionalized with various ligands and peptides. This functionalization of nanoparticles can lead to a multitude of applications. Scientists from a variety of research areas have thus been drawn to NPs because of their rich, unique characteristics.

Nanosystems acting synergistically with cellular components have the potential to integrate diagnostic and therapeutic modalities, like cancer selectivity and bioimaging combined with targeted multimodal cancer therapies [5,6]. Nucleic acid conjugated nanomaterials (NACs) have proven to be a useful tool in cancer research [7,8,9,10,11]. They have the benefit of working as endogenous biomarkers with patient genetic information [12,13]. Graphene oxide/gold hybrid nanostructures (Au@GO NP-NACs) have emerged as an excellent platform for cancer diagnostics, as depicted in (Figure 1). Here, Au@GO NP-NAC nanosystems have been employed for site-specific gene/drug delivery, multimodal treatment, and noninvasive imaging, as illustrated in Figure 1a. Au@GO NP-NACs with unique surface-enhanced Raman spectroscopy (SERS) characteristics were created, and are shown in Figure 1b, for the noninvasive detection of oncogenes. This hybrid method uses Raman dye (Cy5)-labeled nucleic acid for a synergistic decrease in cancer treatment resistance. As shown in Figure 1, synergistic multimodal cancer diagnostics and gene suppression are potential mechanisms for cancer apoptosis (c). Silica-coated Au nanorods showed promise for cancer applications in recent research [14].

The promise of nanotechnology goes beyond the applications in medicine and encompasses all aspects of human life. Here, we would like to mention parenthetically that the United Nations (UN) 2030 Agenda for Sustainable Development Goals (SDGs) was structured around four primary areas: social, economic, environmental, and governance areas [16,17]. The social pillar seeks to address issues such as poverty, poor nutrition, insufficient healthcare, the lack of access to safe water and sanitation, and insufficient sources of renewable energy. The economic pillar deals with the issues of sustainable urban development, innovation, infrastructure, and responsible manufacturing and consumption. The environmental pillar deals with our planet’s climate and marine and land life. Lastly, the governance pillar concerns itself with fostering peace, strong institutions, and partnerships. The UN has mandated the need to pool resources from many sources to achieve these SDGs [18]. In 2019, the UN Department of Economic and Social Affairs released a report titled, “The Future is Now—Science for Achieving Sustainable Development” [19]. Science, technology, and innovation are highlighted as key instruments for societal, healthcare, and economic advancement, and it is emphasized that new technology deployment should be prioritized [19].

Several studies in recent years have demonstrated the potential of nanomaterials to improve human health, water purification, and resource conservation. There have been numerous applications of metal-based nanostructures in the fields of electronics, gene therapy, drug delivery, and environmental remediation. These nanostructures offer exceptional properties, such as a higher surface area, surface energy, and chemical reactivity compared to macroscopic materials [20].

Furthermore, the improved optical and catalytic characteristics, among others, of bimetallic nanoparticles (BMNPs) relative to monometallic nanostructures have recently been investigated [21,22]. It has been determined that BMNPs can contribute to the achievement of several of the UN SDGs, such as excellent health and wellbeing, clean water and sanitation, and responsible consumption and production, with a focus on problems and solutions [23,24,25]. While there is research that suggests that metallic nanoparticles could be useful in Sustainability Agenda initiatives, no reviews exist that establish a direct correlation between these nanostructures and the UN SDGs [26,27,28,29].

The present review outlines the green synthesis trends in BMNPs that align with sustainability initiatives. It takes into consideration the enhanced properties of BMNPs in a wide variety of applications, the availability of nanomaterial-based products in the market over the past decade, the need for more sustainable synthesis methods, and the potential risks and hazards that these nanomaterials may cause in the long run. Depending on their structure, potential chemical composition, and method of synthesis, we highlight the pros and cons of the NPs by describing their properties and traits. To further demonstrate their possible use in connection to the aforementioned SDGs, we present case studies of bimetallic nanostructures produced using both conventional and environmentally friendly synthesis methods, drawing attention to the benefits and practical consequences of the latter. As a conclusion, we discuss the market situation of metallic NPs and their implications for the sustainability agenda, as well as the significance of the synthesis technique in relation to the anticipated applications.

## 2. Properties of Metallic Nanoparticles

Metallic nanostructures exhibit improved features, one of which is surface plasmonic behavior [30]. For instance, the size and structure of AgNPs determine how they interact with light [30]. The localized surface plasmon resonance (LSPR) is created when the conduction electrons surrounding the NP fluctuate in an orderly fashion due to the incident wavelength of light [31]. The development of nanostructured systems for detecting chemical and biological compounds was facilitated using this phenomenon for the electromagnetic augmentation of spectroscopic signals, such as surface-enhanced Raman scattering (SERS) [30]. The biomedical field has potential use for Ag and Au NPs beyond detection, particularly in antibacterial activities related to anticancer treatments [32]. The findings from the study by Soliman et al. [32] demonstrated that silver and gold nanoparticles have potential as antioxidant, anticancer, antibacterial, and antimicrobial agents.

An improved performance in the degradation of environmental toxins can be achieved using metallic nanostructures, which are famous for their catalytic activity [33]. Since copper (Cu) is both abundant and inexpensive, Cu NPs have shown promise as a material for degrading water contaminants such as methylene blue (MB) and Congo red (CR) [34]. In comparison to NaBH4, a commonly used reducing agent, Cu NPs have demonstrated superior catalytic characteristics in the reduction mechanism of the pollutants, resulting in the rapid and thorough elimination of CR and MB [34]. Along with their degrading capabilities, Cu NPs can be used electrochemically, allowing for the rapid and thorough elimination of these pollutants [34]. In addition, Cu NPs have been used electrochemically to diminish CO_2_-saturated aqueous solutions at room temperature and pressure [35].

### 2.1. Comparison between Mono- and Bimetallic Nanoparticles

Materials that combine two distinct metals using bimetallization techniques have improved and altered properties compared to those of a monometallic system. When contrasted with monometallic alternatives, BMNPs exhibit notable benefits [28]. The first benefit of bimetallization is an improvement in the system’s catalytic characteristics over those of monometallic nanoparticles (MNPs) [22]. For example, Pd-Ni, Pt-Ag, and Pd-Au nanowires (NWs) are bimetallic systems that enable electronic transitions within the NWs. These NWs have shown excellent electrochemical performance due to their greater surface area and acceptable stability [26,36]. Furthermore, when combinations of electrical, mechanical, functional, and structural modifications are introduced, they can cause synergistic effects when two metals are present [22].

BMNPs are more functional and have a wider range of applications than MNPs due to the regulated optical, electrical, plasmonic, thermal, and magnetic properties triggered by these interactions [37,38]. The thermal characteristics of a conductive ink containing BMNPs were studied by Yang et al. [34], who used differential scanning calorimetry (DSC) to determine the decomposition temperature of the ink. Adding Cu/Ag NPs to ink raises its decomposition temperature by around 30 °C compared to using just Ag NPs. In addition, it was demonstrated that the Cu:Ag weight ratio may be adjusted in a variable way to manage thermal stability. Anjo and colleagues [39] used a comparable strategy, testing the plasmonic and magnetic characteristics of Fe/Ag NPs produced via laser ablation. Varying the Fe:Ag composition ratio produced UV-Vis spectra that differed significantly, similar to Yang’s study [34]. Due to changes in particle shape and the overall composition of the system, the maximum absorbance wavelength was gradually shifted to the red end of the spectrum as the Fe:Ag ratio increased. All generated samples exhibited superparamagnetic behavior when tested for magnetic properties. However, the system with the optimal magnetic properties was the one that corresponded to a composition ratio of Fe50:Ag50 [39]. Additionally, Malik and colleagues reported on an Fe-Ag nanosystem, where the combination of Fe NPs’ reduction capacity and Ag NPs’ catalytic activity could result in the catalytic reduction of pollutants such as nitroaromatic chemicals [40]. Thus, systems based on noble metals can be systematically improved and used to create inexpensive solutions with improved thermal, plasmonic, magnetic, and catalytic capabilities by including transition metals, such as Cu and Fe. Finally, they are chemically stable and functionalize with ease [41]. Nanobranched bimetallic structures, such as AuCu, can improve the lower limit of detection (LOD) of biomarker detectors, such as glucose ones, with superior selectivity and stability [42]. It is believed that BMNPs could represent a more efficient way to achieve sustainability agenda goals because of these improved attributes.

### 2.2. Structure and Characteristics of Bimetallic Nanoparticles

The classification of bimetallic systems is heavily influenced by their structure, architecture, and composition [43]. Noble metal BMNPs have high catalytic characteristics and strong plasmon resonances because of their electronic configuration, which is determined by their composition [44]. Iron (Fe), nickel (Ni), gold (Au), cobalt (Co), and silver (Ag) are the most popular metals utilized in bimetallic systems.

The composition of noble metal BMNPs determines their electronic configuration, which, in turn, determines their high catalytic properties and allows the display of strong plasmon resonances [44]. As mentioned above, among the most popular metals utilized in bimetallic systems are iron (Fe), nickel (Ni), gold (Au), and cobalt (Co) [44]. The outermost n-electrons and unsaturated (n − 1) d electron shell of these metals contribute to their catalytic and magnetic characteristics, respectively [45]. Finally, since noble metals are expensive due to their low abundance and dispersed distribution in the Earth’s crust [46], combining them with transition metals makes BMNPs more affordable.

## 3. Nanoalloy Synergistic Applications

### 3.1. Gold Nanoalloys

Noble metal nanoparticles have outstanding physicochemical characteristics and are suitable for biomedical applications [47,48]. Both silver (Ag) and gold (Au) nanoparticles have applications in biology, health, and biochemistry [49,50,51]. The antibacterial activity of Ag nanoparticles is well known and used in biomedical and consumer products [52,53,54,55,56,57]. The toxicity of Ag nanoparticles, however, is a major concern in the biomedical field [58,59]. For this reason, Au nanoparticles are preferable due to their biocompatible nature [60] and have been applied to drug delivery, biological imaging, and cancer therapy [61,62,63,64,65,66,67,68,69]. Hybrid (Cu and Au) tripod nanocrystals have been examined both empirically and theoretically by our research group; upon NIR laser irradiation, we discovered a distinct photothermal-effect-based anticancer treatment [6].

In principle, a hybrid Au and Ag nanoalloy can enhance biomedical properties, such as Ag NP toxicity toward bacteria or cells and Au NP biocompatibility properties [70]. A variety of techniques to manufacture Au–Ag nanoalloys have been studied. UV irradiation, sol–gel techniques, and wet chemical synthesis are examples of synthetic procedures [71,72,73,74,75,76]. So far, laser irradiation has been employed for the synthesis of Au–Ag nanoalloys [77]. It has also been reported that the ultrasonication of individual Ag and Au nanoparticles [78] or refluxing with oleylamine [79,80] may produce Au shell/Ag core nanoalloys.

According to Georgios A. Sotiriou et al. [81], surface oxidation and the leaching of Ag ions can be avoided through the presence of Au in the production of Ag nanoparticles. Compared to pure Au nanoparticles, inexpensive and biocompatible Au–Ag nanoalloys have outstanding plasmonic characteristics. Nanoalloys are becoming attractive options in medical sectors because of their strong plasmonic characteristics. Figure 2a depicts the release profile of Ag ions from Ag–Au nanoalloys. The 17 mg L1 Ag ions produced by pure Ag nanoparticles account for around 17% of the overall mass. The discharge of Ag ions increases as the Au concentration decreases. The differential plasmon peak observed in water and ethanol is shown in Figure 2b. The plasmon peak position difference increases as the Au concentration decreases, making it safer (Figure 2c).

Because of its outstanding optical characteristics and high photothermal transition efficiency, the plasmon resonance of metallic nanoparticles plays an important role in disease diagnostics, biological sensing [82], and other medicinal sectors [3]. Various research studies have focused on Ag and Au in nanoalloys [83,84]. However, under typical experimental conditions, the surface of a Ag-nanoparticle can be reduced [85], reducing its plasmonic activity [86]. Surface oxidation reduced by Au-Ag nanoalloys is useful for SERS applications [87,88,89,90]. As a result, the Au-Ag nanoalloy system is a promising platform to produce non-toxic and low-cost plasmonic nanomaterials.

### 3.2. Magnetic Nanoalloys

Magnetic nanoparticles provide an excellent opportunity for hyperthermia ablation and magnetic tumor separation, and act as contrast enhancers in magnetic resonance imaging (MRI) [91,92,93,94,95,96,97,98,99,100,101]. This is a noninvasive, powerful biological imaging technique, that relies on protons in lipids and water molecules for its signal. Magnetic nanoparticles can cause a large contrast between diseased and normal organ tissue in biological imaging, acting as MRI image enhancing agents [102,103]. Magnetic nanoparticles (FeCo, MnFe_2_O_4_, Fe_2_O_3_, Fe_3_O_4_) are common enhancing agents used in MRI imaging of the liver, spleen, gastrointestinal system, and bone marrow [104,105]. Apart from its widespread use, magnetic iron oxide has several drawbacks, including rapid clearance by phagocytic cells and the prevention of tissue penetration and trans-endothelial transit [106]. When targeting an organ, it is critical to produce an MRI contrast agent with increased selectivity, tissue delineation, and longer intravascular retention.

Magnetic nanoalloys have been shown to have longer blood circulation times, improving the possibility of interaction with specific tissues [107]. The biocompatible and inert shell afforded by silica is considered a popular coating material for magnetic nanoalloys. The silica coating may preserve chemicals from deterioration and can facilitate the adorning of the NP surface with functional chemical groups [108,109,110]. Surfaces containing chemical groups provide several advantages, such as solubility and stability in an aqueous medium for biomedical applications [111]. This improves the binding ability of biological compounds and the effectiveness of cell internalization for the targeted administration of medicines and MRI applications [112,113].

Platinum–iron (Pt-Fe) nanoalloys offer a unique form of magnetic nanomaterial because of their high magneto-crystalline anisotropy and Curie temperature [114]. Various synthesis techniques have been investigated in the past to generate platinum iron nanoalloys with different sizes, shapes, and stoichiometries. The co-reduction of platinum and iron materials via low-pressure emulsion protocols [41,115,116,117,118], the high-pressure polyol route, and the photothermal technique are used for the synthesis of magnetic nanoalloys [38,119,120,121,122,123,124].

Medical applications for magnetic nanoalloys have been studied, including T2 MRI contrast agents [125,126] and hyperthermia ablation [127,128], as well as magnetic separation [129,130]. However, NP toxicity remains a serious issue that prevents their widespread use in biological treatment and diagnostics [131,132]. Due to the retention effect and increased permeability, Fe_3_O_4_@Au NPs resulted in high tumor accumulation. As shown in Figure 3, Fe_3_O_4_@Au NPs were utilized for NIR laser-induced photothermal synergistic treatment and multimodal imaging systems [126].

## 4. Mechanism of Nanoparticle–Cell Interaction

In 1857, Michael Faraday studied the preparation and properties of colloidal suspensions of “Ruby” gold, ref. [133] considered to be the first experiment on nanoparticles in modern times. Today, there are a wide range of applications for nanoparticles [134,135,136,137,138,139,140,141,142,143]. Metallic nanoparticles, in contrast to their bulk counterparts, show defined activities that are useful in a variety of biomedical applications. High dangling bonds, electron storage capacity [144], the presence of edges and corners, a high surface energy, surface plasmon resonances (SPR), and a high surface area-to-volume ratio are among the functions [145,146]. Different processes, such as biological, physical, and chemical techniques, can create metallic nanoparticles with diverse shapes and sizes [147,148,149].

Nanoscience has produced different nanomaterials with distinct ligands and groups to display dissimilar functionalities and characteristics. Nanomaterials, because of their tiny size, might effectively react with the biological cell membrane and intracellular fluid [150]. Nanoparticles are most used in antimicrobial processes, catalysis, cancer treatment, imaging, diagnostics, and medication delivery [151,152,153]. Nanoparticles may be partially embedded in cell membranes, exposing them to both extracellular fluid and cytosol. This structure mimics that of a transmembrane protein. Using nanoparticles containing alkyl groups, researchers showed the existence of embedment structures [154]. However, if large numbers of nanoparticles are delivered inside a cell, they may damage subcellular organelles and the cellular membrane itself, thus causing cytotoxicity.

Embedment, inner attachment, free translocation, and the outside wrap are four different ways for the cell membrane and nanoparticles to interact (Figure 4). In terms of the outer wrap, nanoparticles only adhere to the cell membrane’s exterior surface and cannot permeate even a small portion of the membrane. When nanoparticles are large and adorned with a low charge, this structure can be seen (Figure 4). The plasma membrane of the cells contains microdomains that are enriched in certain cholesterol, gangliosides, and glycosphingolipids that form membrane/lipid rafts. Membrane/lipid rafts have myriad functions, including the regulation of cellular polarity and the organization of sorting and trafficking mechanisms. These rafts are also important for forming platforms for intracellular cytoskeletal binding and extracellular matrix adhesion to the plasma membrane. Furthermore, they are involved in the generation of signaling events and constitute the sites where nanoparticles enter the cells.

Nanoparticle interactions with the cellular membrane are achieved through key physicochemical properties such as size, surface charge, and ligand chemistry [154]. Nanoparticles with a diameter less than 1 nm, such as ions or water molecules, enter the cellular membrane via permeation. Larger-size nanoparticles enter the cell membrane through endocytosis, which includes receptor-mediated endocytosis, pinocytosis, and phagocytosis. Medium-size nanoparticles lie on the boundary between endocytosis and permeation. Proteins (3–20 nm) go through complex interaction with cell membranes. Lin et al. [154] found nanoparticles with inner attachment to cell membranes, which results in nanoparticles being exposed to the intracellular fluid as they are attached to the inner surface of the cell membrane via focal adhesion [154]. Only when nanoparticles are coated with hydrophobic groups and for certain size–charge combinations can this structure be seen [154]. Their suspension is then available within the intracellular fluid for free translocation. This situation can arise when nanoparticles are tiny and adorned with a high charge [154].

## 5. Green Synthesis Trends in Bimetallic Nanoparticles

Combining the principles of green chemistry with those of traditional physicochemical methods, we can develop biologically based synthesis processes that are both safe and environmentally friendly and incorporate nanomaterials into industrial processes on a large scale, while reducing the likelihood of harm to humans and the environment [155]. The synthesis of BMNPs in an eco-friendly way must adhere to green chemistry techniques: using solvents and reducing agents that are not harmful to the environment and capping agents that are not poisonous [156]. The field of green nanotechnology came into being when these concepts of green chemistry were applied to the production of nanomaterials [157]. Therefore, BMNPs synthesized and applied using green nanotechnology in sustainability science have demonstrated efficacy against water and soil contamination and antibacterial activity against pathogen bacterial strains in the food and health industries.

Biomolecules derived from many sources, including plants, algae, bacteria, fungi, and organic waste products, such as fruit peels, are commonly utilized as agents for reduction and stabilization [158]. Metallic salts can be reduced in either an intracellular or extracellular environment in bacterial- and fungal-mediated production. Biomedical and technical applications can benefit from zero-valence metallic structures that can be produced through the bacteria-mediated green synthesis of BMNPs. Fungus-mediated syntheses are preferable over bacterium-mediated ones because they can be used in speedier, scaled-up processes [159,160]. Metal NP syntheses mediated by plants are among the most popular methods. It is common practice to begin these procedures by obtaining a plant extract in liquid form; this yields a fine powder that needs to be washed and dried before synthesis [161]. Because plant leaves include various functional groups and phyto-chemicals that work as reducing and stabilizing agents, these interactions are beneficial [162]. For the biomolecule-mediated production of metallic NPs, typical ingredients include poly-ethylene glycol [163] and starch [164]. Finally, one of the less investigated forms of synthesis is that which is mediated by waste materials.

## 6. Conclusions

In shaping the translocation outcome, the physiochemical properties of the nanoparticles along with their synergistic effects play a vital role. Synergistically, they trigger initial pore nucleation for NP translocation. The impact of nanoparticles acting synergistically promise to advance Green Chemistry and Sustainability, as required by the UN sustainability agenda. The study herein reviewed the literature on the synergistic impact of nanoparticles on green nanotechnology. The synergistic effect between the nanocomposite, nanoalloy, and nanoparticles can be conceptualized theoretically. The interactions between various nanoalloys, nanocomposites, and nanoparticles have been discussed. However, the details of the underlying processes need further investigation, as the field of green nanotechnology is still in its infancy. The focus of the scientific community on the use of nanotechnology for technical and consumer applications and in medical diagnostics and treatment during the last decades has resulted in the establishment of a substantial database. We need to further deepen our knowledge on the interaction of various nanoparticles with specific biological barriers and compartments in optimizing internalization and site-specific drug release. 

## Figures and Tables

**Figure 1 ijms-25-03266-f001:**
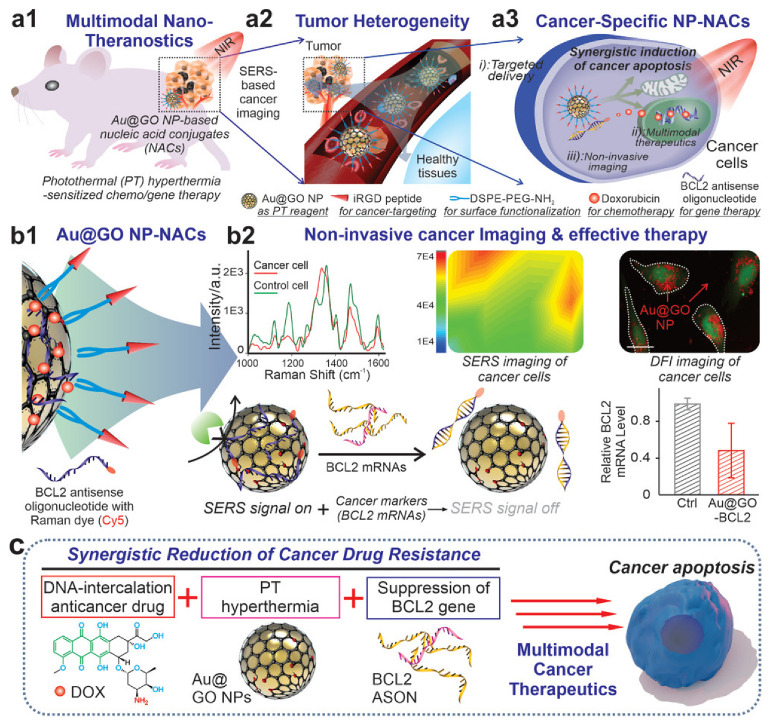
(**a**) The use of Au@GO NP-NACs showed photothermal hyperthermia-enhanced chemo/gene therapy for multimodal therapies. (**b**) For drug resistance, SERS imaging and gene therapy are employed. (**c**) The mechanism postulated for cancer apoptosis in synergistic multimodal cancer theragnostics. With permission, this figure was derived from reference [15].

**Figure 2 ijms-25-03266-f002:**
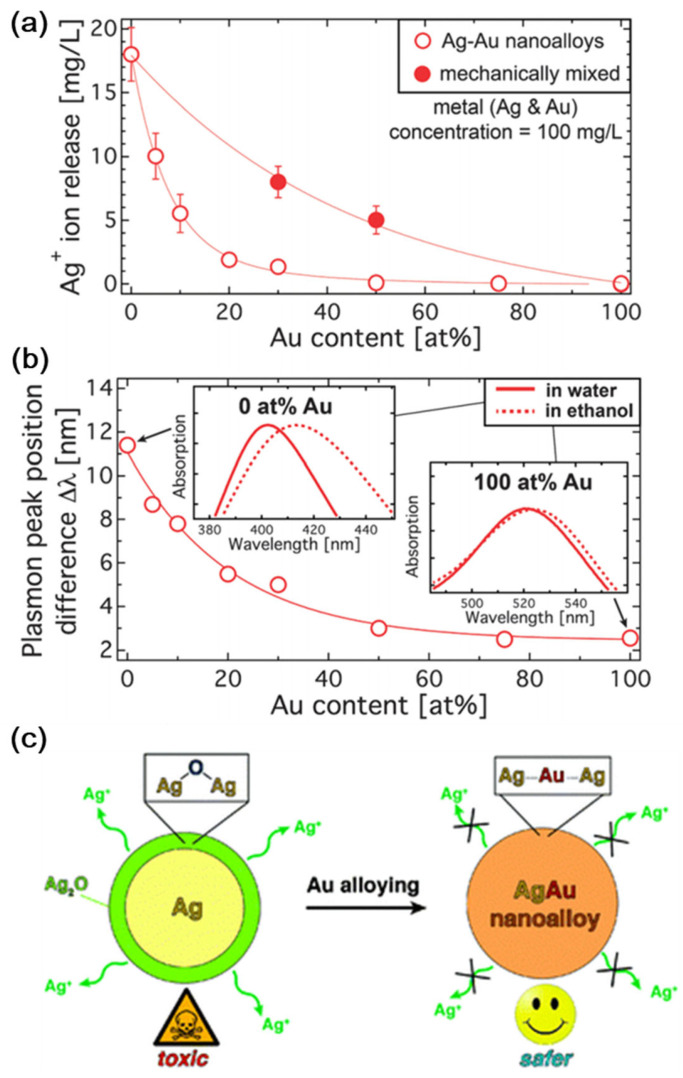
(**a**) The release profile of Ag ions in aqueous solution. (**b**) Plasmon peak of nanoalloys observed in ethanol and water medium. (**c**) Au addition reduces surface oxidation and hazardous Ag+ ion leaching. With permission, this figure was taken from reference [81].

**Figure 3 ijms-25-03266-f003:**
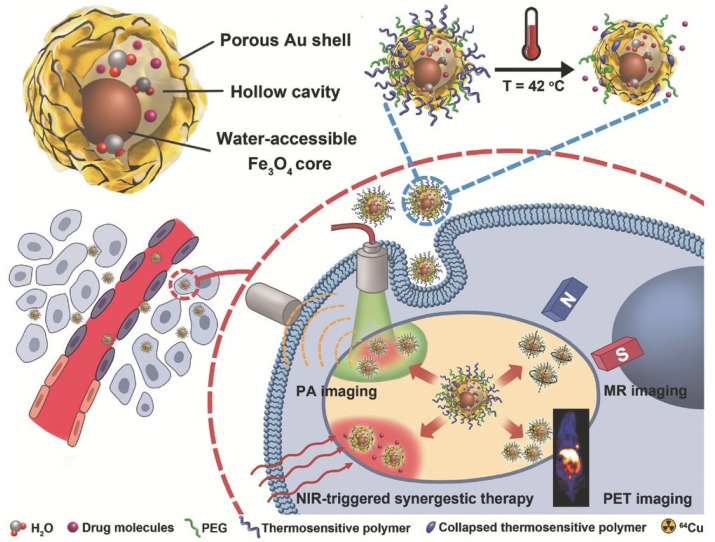
Fe_3_O_4_@Au NPs for NIR-triggered photothermal synergistic treatment and MR/PA/PET multimodal imaging are shown. With permission, this figure was derived from ref. [126].

**Figure 4 ijms-25-03266-f004:**
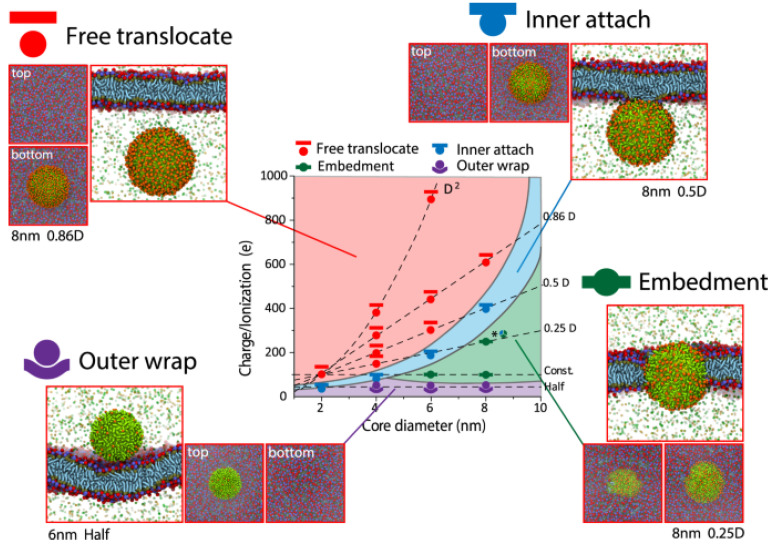
Hydrophobic ligands adorn nanoparticles with various surface ionization/charge and core sizes (alkyls). With permission, this figure was taken from ref. [154].

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
