# Peer review of "Recent Advances in Synergistic Effect of Nanoparticles and Its Biomedical Application"

_ijms, 2024, doi:10.3390/ijms25063266_

Round 1
Reviewer 1 Report
Comments and Suggestions for Authors
Overall comments:
The review entitled ‘Recent Advances in Synergistic Effect of Nanoparticles and its Biomedical Application’ has been peer-reviewed. The authors reviewed the critical role of nanomaterials' synergistic impact in novel drug delivery systems, emphasizing subcellular environments and intracellular delivery with minimal toxicity. They also discussed cell-nanoparticle interactions in terms of translocation, outer wrapping, embedding, interior attachment, and the influence of nanoparticle characteristics such as form, size, surface area, ligand chemistry, and charge. The review was well-written and can be published after following minor concerns are revised.
Minor concerns:
1. Strengthen the Abstract. The authors should clarify the definition of the 'synergistic effect' of nanoparticles, outline the structure of the subsections focusing on discussed nanomaterials, and specify the biomedical applications covered in the article, such as bioimaging, tissue engineering, immunotherapy, and phototherapy.
2. In section 6, it is advisable to address concerns about cytotoxicity according to the interaction mechanisms. This should include an examination of how surface functional groups and surface charge impact cytotoxicity.
3. In the Conclusion section, the authors can present future outlooks for translating nanoparticles into clinical settings. This should encompass discussions on organ distribution, clearance pathways, regulatory considerations, storage issues, and other relevant aspects.
4. Verify that all full forms of the abbreviations in the reproduced figures are noted in the manuscript. If any are missing, include them in the respective figure captions.
Comments on the Quality of English Languagenone
Author Response
Reviewer 1:
Overall comments:
The review entitled ‘Recent Advances in Synergistic Effect of Nanoparticles and its Biomedical Application’ has been peer-reviewed. The authors reviewed the critical role of nanomaterials' synergistic impact in novel drug delivery systems, emphasizing subcellular environments and intracellular delivery with minimal toxicity. They also discussed cell-nanoparticle interactions in terms of translocation, outer wrapping, embedding, interior attachment, and the influence of nanoparticle characteristics such as form, size, surface area, ligand chemistry, and charge. The review was well-written and can be published after following minor concerns are revised.
Minor concerns:
- Strengthen the Abstract. The authors should clarify the definition of the 'synergistic effect' of nanoparticles, outline the structure of the subsections focusing on discussed nanomaterials, and specify the biomedical applications covered in the article, such as bioimaging, tissue engineering, immunotherapy, and phototherapy.
Answer: Thank you. We have revised the abstract section on page 2 of the marked version.
After revision:
The synergistic impact of nanomaterials is critical for novel intracellular and/or subcellular drug delivery systems of minimal toxicity. This synergism results in a fundamental bio/nano interface interaction, which is discussed in terms of nanoparticle translocation, outer wrapping, embedding, and interior cellular attachment. The morphology, size, surface area, ligand chemistry and charge of nanoparticles, all play a role in translocation. In this review we suggest a generalized mechanism to characterize the bio/nano interface, as we discuss the synergistic interaction between nanoparticles and cells, tissues, and other biological systems. Novel perceptions are reviewed regarding the ability of nanoparticles to improve hybrid nanocarriers with homogeneous structure to enhance multifunctional biomedical applications such as bioimaging, tissue engineering, immunotherapy, and phototherapy.
- In section 6, it is advisable to address concerns about cytotoxicity according to the interaction mechanisms. This should include an examination of how surface functional groups and surface charge impact cytotoxicity.
Answer: Thank you. We mentioned it on page 17-18 of the marked version.
After revision:
Nanoparticle interaction with cellular membrane is achieved through key physicochemical properties such as size, surface charge and ligand chemistry [167]. Nanoparticles with a diameter less than 1 nm, such as ions or water molecules, enter the cellular membrane via permeation. Larger size nanoparticles enter the cell membrane through endocytosis, which includes receptor mediated endocytosis, pinocytosis, and phagocytosis. Medium size nanoparticles lie on the boundary between endocytosis and permeation. Proteins (3-20 nm) go through complex interaction with cell membranes. Lin et. al. [167] found nanoparticles with inner attachment to cell membranes, which results in nanoparticles being exposed to the intracellular fluid as they are attached to the inner surface of the cell membrane by focal adhesion [167]. Only when nanoparticles are coated with hydrophobic groups and for certain size-charge combinations can this structure be seen [167]. Their suspension is then available within the intracellular fluid for free translocation. This situation can arise when nanoparticles are tiny and adorned with a high charge [167].
- In the Conclusion section, the authors can present future outlooks for translating nanoparticles into clinical settings. This should encompass discussions on organ distribution, clearance pathways, regulatory considerations, storage issues, and other relevant aspects.
Answer: Thank you. We mentioned it on page 20 of the marked version.
After revision:
In shaping the translocation outcome, the physiochemical properties of the nanoparticles along with their synergistic effect play a vital role. Synergistically, they trigger initial pore nucleation for NP translocation. The impact of nanoparticles acting synergistically promise to advance Green Chemistry and Sustainability, as required by the UN sustainability agenda. The study herein reviewed the literature on the synergistic impact of nanoparticles on Green Nanotechnology. The synergistic effect between nanocomposite, nanoalloy, and nanoparticles can be conceptualized theoretically. The interactions between various nanoalloys, nanocomposites, and nanoparticles have been discussed. However, the details of the underlying processes need further investigation, as the field of Green Nanotechnology is still in its infancy.
- Verify that all full forms of the abbreviations in the reproduced figures are noted in the manuscript. If any are missing, include them in the respective figure captions.
Answer: Thank you. We have verified all abbreviations.
Reviewer 2 Report
Comments and Suggestions for Authors
This manuscript provides an overview of recent developments related to synergistic effects of nanoparticles, with a focus on nanoalloys and nanocomposites. While the topic is interesting and timely, I have some major concerns that should be addressed before this paper can be considered for publication. Major points 1. The introduction needs significant reworking. The goals and scope of the review are unclear. There is a disjointed mix of introductory background, specific examples, and discussion of synergistic effects. I suggest reorganizing this section to clearly state the objectives, scope, and importance of synergistic nanoparticle systems in biomedicine. Start broad, provide context, and build to your specific focus. 2. The discussion of nanoalloys jumps between different metal combinations and applications without a clear flow or organization. Consider grouping nanoalloys by composition (e.g. separate sections for Au, Ag, Fe, etc.) and providing more details on their synergistic effects in each biomedical application. 3. More detail is needed in explaining the mechanisms and origins of synergistic effects in nanoalloys and nanocomposites. How exactly does combining multiple nanomaterials lead to improved properties and performance? Elaborate on the chemical, physical, and structural factors underlying synergism. 4. The section 5 (Nanocomposite synergistic application) on nanocomposites and biosurfactants seems unfinished and disconnected from the rest of the review. Either integrate it better with the synergistic theme or consider removing it to sharpen the focus on nanoalloys. 5. The review would benefit from summary tables comparing the properties, advantages, synthesis methods, etc. of different nanoalloy compositions and architectures. Visual abstracts or graphical illustrations of synergistic mechanisms would also help readers. 7. The conclusion is underdeveloped and does not summarize the key points or future outlook based on the current state of synergistic nanoparticle research. Please expand the conclusion to tie the review together.
Comments on the Quality of English LanguageExtensive editing of English language required
Reviewer 3 Report
Comments and Suggestions for Authors
I find the title of this review (Recent advances in syntegitic effect of nanoparticles and its biomedical applications) appealing a interesting. However, after reading the differente sections I found that no new information or a enriching discussion on the topics contained in it were offered by the authors. The review seems to be a plain compilation of information on some weakly related topics, and it lacks of a deep analysis to understand and rationalize the origin of these so-called synergistic interactions for potential biomedical uses. There are also some minor (but significative) mistakes (for example, "FeCo" nanoparticles are a magnetic alloy, but "FeCO" is not correct). At its present form, this review lacks of a deep analysis and discussion on the title-planted topic.
Comments on the Quality of English LanguageEnglish is good and readable.
Author Response
I find the title of this review (Recent advances in syntegitic effect of nanoparticles and its biomedical applications) appealing a interesting. However, after reading the differente sections I found that no new information or a enriching discussion on the topics contained in it were offered by the authors. The review seems to be a plain compilation of information on some weakly related topics, and it lacks of a deep analysis to understand and rationalize the origin of these so-called synergistic interactions for potential biomedical uses. There are also some minor (but significative) mistakes (for example, "FeCo" nanoparticles are a magnetic alloy, but "FeCO" is not correct). At its present form, this review lacks of a deep analysis and discussion on the title-planted topic.
Answer: Thank you. We have carefully considered the comments and revised the whole manuscript.
Round 2
Reviewer 2 Report
Comments and Suggestions for Authors
It is commendable to note that the authors have made significant efforts in enhancing the clarity and depth of their review on the synergistic effects of nanoparticles in biomedical applications. Here are the suggestions for further improvement:
1. A more detailed exploration of the underlying mechanisms of the synergistic effects of nanoparticles could greatly enhance the manuscript. Providing a deeper understanding of the interactions at the molecular and cellular levels would be invaluable for readers seeking to grasp the complexities of this field.
2. The conclusion effectively summarizes the paper but could be strengthened by offering a perspective on future research directions. Highlighting emerging trends, potential breakthroughs, and unexplored avenues could inspire further investigation and innovation in this exciting area of research.
Considering the comprehensive review provided, the insightful discussions, and the manuscript's contribution to the field, I recommend acceptance after minor revisions. Addressing the areas for improvement outlined above will undoubtedly enhance the manuscript's impact.
Comments on the Quality of English LanguageMinor editing of English language required
Author Response
Reviewer -2:
It is commendable to note that the authors have made significant efforts in enhancing the clarity and depth of their review on the synergistic effects of nanoparticles in biomedical applications. Here are the suggestions for further improvement:
We appreciate you for your precious time in reviewing our paper and providing valuable comments. It was your valuable and insightful comments that led to possible improvements in the current version. We have carefully considered the comments and tried our best to address every one of them. We hope the manuscript after careful revisions meets your high standards.
- A more detailed exploration of the underlying mechanisms of the synergistic effects of nanoparticles could greatly enhance the manuscript. Providing a deeper understanding of the interactions at the molecular and cellular levels would be invaluable for readers seeking to grasp the complexities of this field.
Answer: Thank you. We have added it to page 12.
After revision:
The plasma membrane of the cells contains microdomains that are enriched in certain cholesterol, gangliosides, and glycosphingolipids that form membrane/lipid rafts. Membrane/lipid rafts have myriad functions including the regulation of cellular polarity and organization of sorting and trafficking mechanisms. These rafts are also important for forming platforms for intracellular cytoskeletal binding and extracellular matrix adhesion to the plasma membrane. Furthermore, they are involved in the generation of signaling events and constitute the sites where nanoparticles enter the cells.
- The conclusion effectively summarizes the paper but could be strengthened by offering a perspective on future research directions. Highlighting emerging trends, potential breakthroughs, and unexplored avenues could inspire further investigation and innovation in this exciting area of research.
Answer: Thank you. We have added it to page 14.
After revision:
The focus of the scientific community on the use of nanotechnology for technical and consumer applications and in medical diagnostic and treatment during the last decades has resulted in establishing a substantial database. We need to further deepen our knowledge on the interaction of various nanoparticles with specific biological barriers and compartments in optimizing internalization and site-specific drug release.
Considering the comprehensive review provided, the insightful discussions, and the manuscript's contribution to the field, I recommend acceptance after minor revisions. Addressing the areas for improvement outlined above will undoubtedly enhance the manuscript's impact.
Thank you.
Reviewer 3 Report
Comments and Suggestions for Authors
This review has been properly re-written by the authors, so it could comply with the different comments that previous reviewers indicated. At its present form, it seems to be a little bit more concise, organized and clear. I recommend publication as it is.
Author Response
Reviewer -3:
This review has been properly re-written by the authors, so it could comply with the different comments that previous reviewers indicated. At its present form, it seems to be a little bit more concise, organized and clear. I recommend publication as it is.
Answer: Thank you for your valuable comments.